# Base of RoPE Bounds Context Length

**Mingyu Xu**[1]*, **Xin Men**[1]*, **Bingning Wang**[1]†, **Qingyu Zhang**[2],
**Hongyu Lin**[2], **Yaojie Lu**[2], **Xianpei Han**[2] **and Weipeng Chen**[1]

[1] Baichuan Inc.
[2] Chinese Information Processing Laboratory
Institute of Software, Chinese Academy of Sciences
{menxin,xumingyu,daniel}@baichuan-inc.com
{zhangqingyu2024,hongyu,yaojie,xianpei}@iscas.ac.cn

## Abstract

Position embedding is a core component of current Large Language Models (LLMs). Rotary position embedding (RoPE), a technique that encodes the position information with a rotation matrix, has been the de facto choice for position embedding in many LLMs, such as the Llama series. RoPE has been further utilized to extend long context capability, which is roughly based on adjusting the *base* parameter of RoPE to mitigate out-of-distribution (OOD) problems in position embedding. However, in this paper, we find that LLMs may obtain a superficial long-context ability based on the OOD theory. We revisit the role of RoPE in LLMs and propose a novel property of long-term decay, deriving that the *base of RoPE bounds context length*: there is an absolute lower bound for the base value to obtain certain context length capability. Our work reveals the relationship between context length and RoPE base both theoretically and empirically, which may shed light on future long context training.

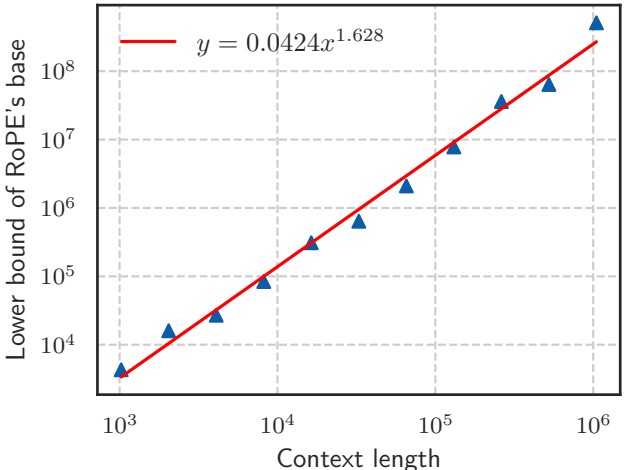

Figure 1: Context length and its corresponding lower bound of RoPE's base value.

---

*Equal contribution. Order determined by swapping the one in [1]
†Corresponding author.

38th Conference on Neural Information Processing Systems (NeurIPS 2024).

# 1 Introduction

In the past few years, large language models have demonstrated surprising capabilities and undergone rapid development. By now, LLMs have been widely applied across various domains, including chatbots, intelligent agents, and code assistants [2, 3]. The Transformer [4], based on the attention mechanism, has been the most popular backbone of LLMs due to its good performance and scaling properties [5]. One of the key component modules in the Transformer is position embedding, which is introduced to embed positional information that is vital for processing sequential data. Rotary position embedding (RoPE), which encodes relative distance information in the form of absolute position embedding [6], has been a popular choice and applied in many LLMs [7, 8, 9].

RoPE introduces no training parameters and shows improvement in language modeling and many other tasks [6, 10]. One reason that RoPE is widely used is its ability for context length extrapolation [11, 12], which extends the context length of a trained LLM without expensive retraining. In practice, many works [7, 13, 14] have successfully extended the window length by simply increasing base value, the only one hyper-parameter in RoPE, and fine-tuning on long texts.

The reasons behind the success of these long context extensions are often explained as avoiding out-of-distribution (OOD) rotation angles [15, 16] in RoPE, meaning the extended context length (OOD) can be mapped to the in-distribution context length that has been properly trained. Based on the OOD theory, a recent study [15] finds that a smaller base can mitigate OOD and is beneficial for the model's ability to process long contexts, which inspires us to further study the relationship between the base of RoPE and the length of context the model can process.

In this paper, we find that the model may show superficial long context capability with an inappropriate RoPE base value, in which case the model can only preserve low perplexity but loses the ability to retrieve long context information. We also show that the out-of-distribution (OOD) theory in position embedding, which motivates most length extrapolation works [11, 12, 15], is insufficient to fully reflect the model's ability to process long contexts. Therefore, we revisit the role of RoPE in LLMs and derive a novel property of long-term decay in RoPE: the ability to pay more attention to similar tokens than random tokens decays as the relative distance increases. While previous long context works often focus on the relative scale of the RoPE base, based on our theory, we derive an absolute lower bound for the base value of RoPE to obtain a certain context length ability, as shown in Figure 1. To verify our theory, we conducted thorough experiments on various LLMs such as Llama2-7B [17], Baichuan2-7B [8] and a 2-billion model we trained from scratch, demonstrating that this lower bound holds not only in the fine-tuning stage but also in the pre-training stage.

We summarize the contributions of the paper as follows:

- **Theoretical perspective**: we derive a novel property of long-term decay in RoPE, indicating the model's ability to attend more to similar tokens than random tokens, which is a new perspective to study the long context capability of the LLMs.

- **Lower Bound of RoPE's Base**: to achieve the expected context length capability, we derive an absolute lower bound for RoPE's base according to our theory. In short, the base of RoPE bounds context length.

- **Superficial Capability**: we reveal that if the RoPE's base is smaller than a lower bound, the model may obtain superficial long context capability, which can preserve low perplexity but lose the ability to retrieve information from long context.

# 2 Background

In this section, we first introduce the Transformer and RoPE, which are most commonly used in current LLMs. Then we discuss long context methods based on the OOD of rotation angle theory.

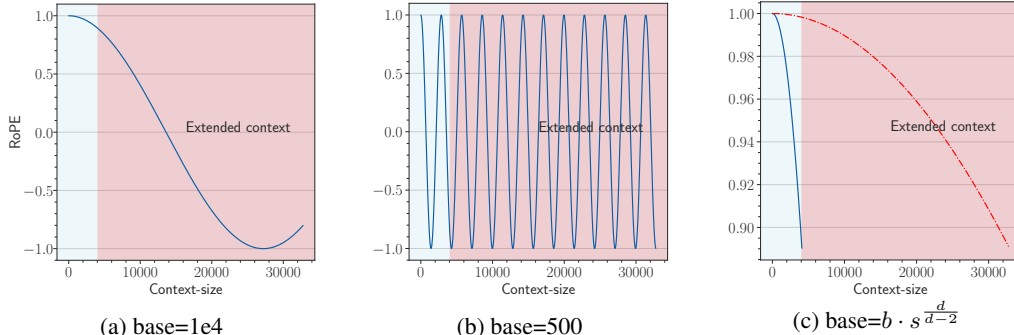

| (a) base=1e4 | (b) base=500 | (c) base=$b \cdot s^{\frac{d}{d-2}}$ |

Figure 2: An illustration of OOD in RoPE when we extend context length from 4k to 32k, and two solutions to avoid the OOD. We show the last dimension as it is the lowest frequency part of RoPE, which suffers OOD mostly in extrapolation. (a) For a 4k context-length model with base value as 1e4, when we extend the context length to 32k without changing the base value, the context length from 4k to 32k is OOD for RoPE (red area in the figure). (b) OOD can be avoided with a small base value like 500 [15], since the full period has been fitted during fine-tuning stage. (c) We set base as $b \cdot s^{\frac{d}{d-2}}$ from NTK [11].The blue line denotes the pre-training stage (base=1e4) and the red dashed line denotes the fine-tuning stage (base=$b \cdot s^{\frac{d}{d-2}}$), we can observe that the RoPE's rotation angle of extended positions is in-distribution.

## 2.1 Attention and RoPE

The LLMs in current are primarily based on the Transformer [4]. The core component of it is the calculation of the attention mechanism. The naive attention can be written as:

$$A_{ij} = q_i^T k_j \tag{1}$$

$$\text{ATTN}(X) = \text{softmax}(A/\sqrt{d})\, v, \tag{2}$$

where $A \in R^{L \times L}$ $q, k, v \in R^d$. Position embedding is introduced to use the order of the sequence in attention.

RoPE [6] implements relative position embedding through absolute position embedding, which applies rotation matrix into the calculation of the attention score in Eq. 1, which can be written as:

$$A_{ij} = (R_{i,\theta} q_i)^T (R_{j,\theta} k_i) = q_i^T R_{j-i,\theta} k_j = q_i^T R_{m,\theta} k_j, \tag{3}$$

where $m = j - i$ is the relative distance of $i$ and $j$, $R_{m,\theta}$ is a rotation matrix denoted as:

$$
\begin{bmatrix}
cos(m\theta_0) & -sin(m\theta_0) & 0 & 0 & \cdots & 0 & 0 \\
sin(m\theta_0) & cos(m\theta_0) & 0 & 0 & \cdots & 0 & 0 \\
0 & 0 & cos(m\theta_1) & -sin(m\theta_1) & \cdots & 0 & 0 \\
0 & 0 & sin(m\theta_1) & cos(m\theta_1) & \cdots & 0 & 0 \\
\vdots & \vdots & \vdots & \vdots & \ddots & \vdots & \vdots \\
0 & 0 & 0 & 0 & \cdots & cos(m\theta_{d/2-1}) & -sin(m\theta_{d/2-1}) \\
0 & 0 & 0 & 0 & \cdots & sin(m\theta_{d/2-1}) & cos(m\theta_{d/2-1})
\end{bmatrix}
\tag{4}
$$

Generally, the selection of rotation angles satisfies $\theta_i = base^{-2i/d}$, the typical base value for current LLMs is 10,000.

## 2.2 OOD theory of relative rotation angle

Based on RoPE, researchers have proposed various methods to extend the long context ability of LLMs, among which representatives are PI [12] and NTK-series (NTK-aware [18], YaRN [11], and Dynamical-NTK [19]). Those methods depend on the relative scale $s = T_{\text{new}}/T_{\text{origin}}$, where $T_{\text{origin}}$ is the training length of the original pre-trained model and $T_{\text{new}}$ is the training length in long-context fine-tuning.

**PI** PI directly interpolates the position embedding, and the calculation of $A_{ij}$ becomes:

$$A_{ij} = (R_{i/s} q_i)^T (R_{j/s} k_i) = q_i^T R_{(j-i)/s} k_j = q_i^T R_{m/s} k_j, \tag{5}$$

In other words, the position embedding of the token at position $i$ in pre-training becomes $i/s$ in fine-tuning, ensuring the position embedding range of the longer context remains the same as before.

**NTK-series** The idea is that neural networks are difficult to learn high-frequency features, and direct interpolation can affect the high-frequency parts. Therefore, the NTK-aware method achieves high-frequency extrapolation and low-frequency interpolation by modifying the base value of RoPE. Specifically, it modifies the base $b$ of the RoPE to:

$$b_{\text{new}} = b \, s^{\frac{d}{d-2}}. \tag{6}$$

The derivation of this expression is derived from $T_{\text{new}} b_{\text{new}}^{-\frac{d-2}{d}} = T_{\text{origin}} b^{-\frac{d-2}{d}}$ to ensure that the lowest frequency part being interpolated.

A recent study [15] proposes to set a much smaller base (e.g. 500), in which case $\theta_i = base^{-\frac{2i}{d}}$ is small enough and typical training length (say 4,096) fully covers the period of $\cos(t-s)\theta_i$, so the model can obtain longer context capabilities.

One perspective to explain current extrapolation methods is the OOD of rotation angle [15, 16]. If all possible values of $\cos(t-s)\theta_i$ have been fitted during the pre-training stage, OOD would be avoided when processing longer context. Figure 2 demonstrates how these methods avoid OOD of RoPE.

## 3   Motivation

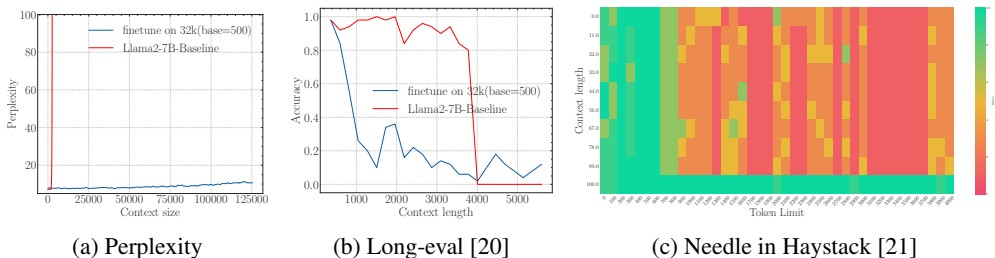

| (a) Perplexity | (b) Long-eval [20] | (c) Needle in Haystack [21] |

Figure 3: The superficial long context capability of avoiding OOD by the smaller base. Following the recent work [15], we fine-tune Llama2-7B with a small base (500) to a context length of 32k.

Recent advancements in long-context language models have seen widespread adoption of NTK-based methods [7, 13, 14]. However, a curious trend has emerged: practitioners often employ significantly larger base values than those originally suggested by NTK-aware approaches. This discrepancy raises critical questions about the efficacy of current theoretical frameworks. Why do practitioners deviate from the recommendations of NTK-based methods? Is the out-of-distribution (OOD) theory underlying these methods insufficient to unlock long-context capabilities fully?

On the other hand, recent research [15], driven by OOD theory, proposes using a much smaller base for RoPE to extend context length. However, our findings, as illustrated in Figure 3, suggest that this approach may only provide superficial long-context capability[22]. While achieving low perplexity even at 128k context length (explicable by OOD theory), the model fails to retrieve relevant information for context lengths as short as 1kwell below its pre-trained length. The observation suggests that the small base determined by OOD theory can't unlock true long-context capability.

These phenomena motivate us to delve deeper into the relationship between RoPE's base and context length. To address the gap between OOD theory and our observations, we conduct a theoretical exploration in the next section, aiming to uncover the underlying mechanisms of effective long-context modeling.

## 4   Theory Perspective

For attention mechanism in language modeling, we have the following desiderata:

**Desiderata 1** *__The closer token gets more attention__: the current token tends to pay more attention to the token that has a smaller relative distance.*

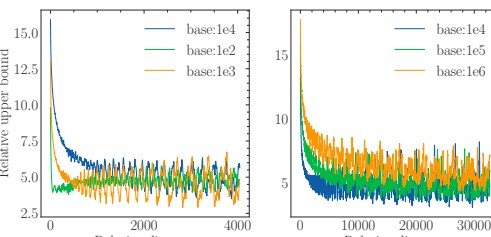
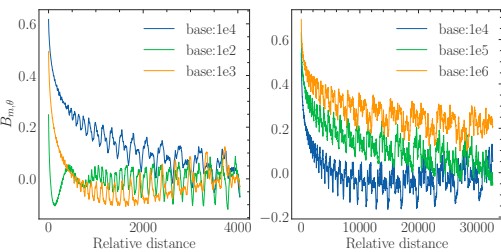

Figure 4: The upper bound of attention score with respect to the relative distance.

Figure 5: The ability to attend more to similar tokens than random tokens.

**Desiderata 2** *The similar token gets more attention: the token tends to pay more attention to the token whose key value is more similar to the query value of the current token.*

Then we examine the desiderata when we apply RoPE to the attention mechanism in LLMs.

### 4.1 Long-term Decay of Upper Bound of Attention Score

For Desiderata 1, the property of RoPE makes the model attend more to closer tokens. This kind of long-term decay has been thoroughly discussed in previous work [6, 23]. It comes from the upper bound of attention score calculation, which can be written as:

$$|A_{ij}| = |q_i^T R_m k_j| \le \max_l(|h_l - h_{l+1}|) \sum_{n=1}^{d/2} |S_n|$$

$$= \max_l(|h_l - h_{l+1}|) \sum_{n=1}^{d/2} |\sum_{l=0}^{n-1} e^{(j-i)\theta_l \sqrt{-1}}|, \tag{7}$$

where $h_l = q_i^T[2l : l2 + 1]k_j[2l : 2l + 1]$. Equation 7 indicates that the upper bound of the attention score $|A_{ij}|$ decays as the relative distance increases. Figure 4 shows the long-term decay curve of this upper bound, which is in accordance with previous findings [6, 23].

### 4.2 Long-term Decay of the Ability to Attend More to Similar Tokens than Random Tokens

In addition to the attention score's upper bound, we also find there exists another long-term decay property in RoPE: the ability to attend more to similar tokens than random tokens decays as the relative distance increases. We define the ability to attend more to similar tokens than random tokens as:

$$\mathbb{E}_{q,k^*}\left[q^T R_{m,\theta} k^*\right] - \mathbb{E}_{q,k}\left[q^T R_{m,\theta} k\right], \tag{8}$$

where $q \in R^d$ is the query vector for the current token, $k^* = q + \epsilon$ is the key value of a similar token, where $\epsilon$ is a small random variable, $k \in R^d$ is the key vector of a random token, $R_{m,\theta}$ is the rotation matrix in RoPE. The first term in Eq. 8 is the attention score of $q$ and a similar token $k^*$, the second term in Eq. 8 is the attention score of $q$ and random token $k$. Then we derive the following theorem:

**Theorem 1** *Assuming that the components of query $q \in R^d$ and key $k \in R^d$ are independent and identically distributed, their standard deviations are denoted as $\sigma \in R$. The key $k^* = q + \epsilon$ is a token similar to the query, where $\epsilon$ is a random variable with a mean of 0. Then we have:*

$$\frac{1}{2\sigma^2}(\mathbb{E}_{q,k^*}\left[q^T R_{m,\theta} k^*\right] - \mathbb{E}_{q,k}\left[q^T R_{m,\theta} k\right]) = \sum_{i=0}^{d/2-1} \cos(m\theta_i) \tag{9}$$

The proof is shown in Appendix A. We denote $\sum_{i=0}^{d/2-1} \cos(m\theta_i)$ as $B_{m,\theta}$, and according to Theorem 1, $B_{m,\theta}$ measures the ability to give more attention to similar tokens than random tokens, which decreases as the relative distance $m$ increases, as shown in Figure 5. For a very small base value, we can observe that the $B_{m,\theta}$ is even below zero at a certain distance, meaning the random tokens have larger attention scores than the similar tokens, which may be problematic for long context modeling.

Table 1: Context length and its corresponding lower bound of RoPE's base.

| Context Len. | 1k | 2k | 4k | 8k | 16k | 32k | 64k | 128k | 256k | 512k | 1M |
|---|---|---|---|---|---|---|---|---|---|---|---|
| Lower Bound | 4.3e3 | 1.6e4 | 2.7e4 | 8.4e4 | 3.1e5 | 6.4e5 | 2.1e6 | 7.8e6 | 3.6e7 | 6.4e7 | 5.1e8 |

### 4.3 Base of RoPE Bounds the Context Length

To satisfy the Desiderata 2, we will get $\mathbb{E}_{q,k^*}\left[q^T R_{m,\theta} k^*\right] \geq \mathbb{E}_{q,k}\left[q^T R_{m,\theta} k\right]$. According to Theorem 1, $B_{m,\theta}$ needs to be larger than zero. Given the $\theta$ in RoPE, the context length $L_\theta$ that can be truly obtained satisfies:

$$L_\theta = \sup\{L | B_{m,\theta} \geq 0, \forall m \in [0, 1, ..., L]\} \tag{10}$$

In other word, if we follow the setting that $\theta_i = base^{-2i/d}$, in order to get the expected context length $L$, there is a lower bound of the base value $base_L$:

$$base_L = \inf\{base | B_{m,\theta} \geq 0, \forall m \in [0, 1, ..., L]\} \tag{11}$$

In summary, the RoPE's base determines the upper bound of context length the model can truly obtain. Although there exists the absolute lower bound, Eq. 9 and Eq. 11 are hard to get the closed-form solution since $B_{m,\theta}$ is a summation of many cosine functions. Therefore, in this paper, we get the numerical solution. Table 1 shows this lower bound for context length ranging from 1,000 to one million. In Figure 1, we plot the context length and corresponding lower bound, we can observe that as the context length increases, the required base also increases.

*Note: this boundary is not very strict because the stacking of layers in LLMs allows the model to extract information beyond the single layers' range, which may increase the context length in Eq. 10 and decrease the base in Eq. 11.* Notwithstanding, in Section 5 we find that the derived bound approximates the real context length in practice.

**Long-term decay from different perspectives.** The long-term decay in section 4.1 and section 4.2 are from different perspectives. The former refers to the long-term decay of the attention score as the relative distance increases. This ensures that current tokens tend to pay more attention to the tokens closer to them. The latter indicates that with the introduction of the rotation matrix in attention, the ability to discriminate the relevant tokens from irrelevant tokens decreases as the relative distance increases. Therefore, a large $B_{m,\theta}$, corresponding to a large base value, is important to keep the model's discrimination ability in long context modeling.

## 5 Experiment

In this section, we conduct thorough experiments. The empirical result can be summarized in Table 2, the details are in the following sections.

Table 2: In Section 5, we aim to answer the following questions.

| Questions | Answers |
|---|---|
| Q: Does RoPE's base bounds the context length during the fine-tuning stage? | Yes. When the base is small, it is difficult to get extrapolation for specific context length. |
| Q: Does RoPE's base bounds the context length during the pre-training stage? | Yes. Our proposed lower bound for RoPE's base also applies to pre-training. If we train a model from scratch with a small base but the context length is large (larger than the bounded length), the resulting model has very limited context length capabilities, meaning some of the context in pre-training is wasted. |
| Q: What happened when base is set smaller than the lower bound? | The model will get the superficial long context capability. The model can keep perplexity low, but can't retrieve useful information from long context. |

### 5.1 Experiments Setup

For fine-tuning, we utilized Llama2-7B [7] and Baichuan2-7B [8], both of which are popular open-source models employing RoPE with a base of $1e4$. We utilized a fixed learning rate of 2e-5 and a

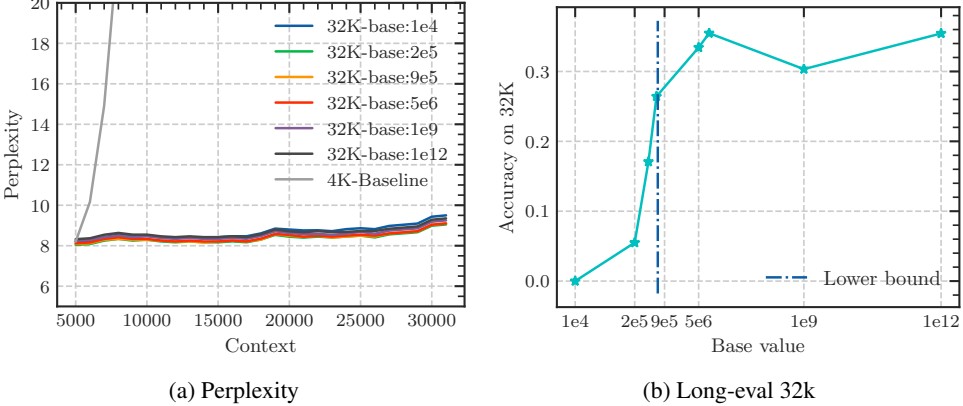

|     |     |
| :-: | :-: |
| (a) Perplexity | (b) Long-eval 32k |

Figure 6: Fine-tuning Llama2-7B-Base on 32k context length with varying RoPE's base. Although the perplexity remains low with varying bases, the Long-eval accuracy reveals a discernible bound for the base value, below which the Long-eval accuracy declines significantly. The dotted line denotes the lower bound derived from Eq. 11 and code is provided in Appendix E

global batch size of 128 and fine-tuning for 1000 steps. For pre-training, we trained a Llama-like 2B model from scratch for a total of 1 trillion tokens. We set the learning rate to 1e-4 and adopted a cosine decay schedule, with models trained on a total of 1T tokens. The dataset we used is a subset of RedPajama [24]. More details of the experimental setup are provided in Appendix B.

Our evaluation focused on two aspects: (1) **Perplexity**: we use PG19 dataset [25] which are often used in long context evaluation; (2) **Retrieval**: in addition to perplexity, we also adopt retrieval since it represents the real long-context understanding ability of LLMs. We choose a) Long-eval benchmark from [20] and b) Needle in a haystack (NIH) [21]. The Long-eval benchmark generates numerous random similar sentences and asks the model to answer questions based on a specific sentence within the context, while the NIH requires the model to retrieve information from various positions in the long context.

## 5.2 Base of RoPE bounds context length in fine-tuning stages

According to Eq. 11, there is a lower bound of RoPE's base determined by expected context length. We fine-tune Llama2-7b-Base on 32k context with varying bases. As depicted in Figure 6, although the difference in perplexity between different bases is negligible, the accuracy of Long-eval varies significantly. In Figure 6b, the dotted line denotes the lower bound derived from Eq. 11, below which the Long-eval accuracy declines significantly. Additional results are provided in Appendix C. Notably, this empirically observed lower bound closely aligns with our theoretical derivation. On the other hand, we can see that $base = 2e5$ achieves the best perplexity, but the accuracy of Long-eval is very low, which indicates the limitations of perplexity in evaluating long context capabilities. We also provide the more comprehensive RULER [26]benchmark results in Appendix G.

## 5.3 The Base of RoPE bounds context length in pre-training stages

According to **Theorem 1** and **Eq. 11**, these constraints could also apply to the pre-training stage. To validate this, we trained a 2B model from scratch with RoPE base=100. The results, depicted in the first row of Figure 7, indicate that even though the model was trained with a context length of 4,096 tokens, it was capable of retrieving information from only the most recent approximately 500 tokens. This demonstrates that the base parameter bounds the context length during the pre-training stage as well. We define the context length from which the model can effectively retrieve information as the effective context length.

According to our theory, the effective context length can be extended as the RoPE's base increases. To validate this, we further fine-tune this 2B model on 32k context length, with RoPE's base set to 1e4, as shown in the second row of Figure 7. While the effective context length increased, it remains significantly below 32k since the effective context length bounded by base=1e4 is much smaller

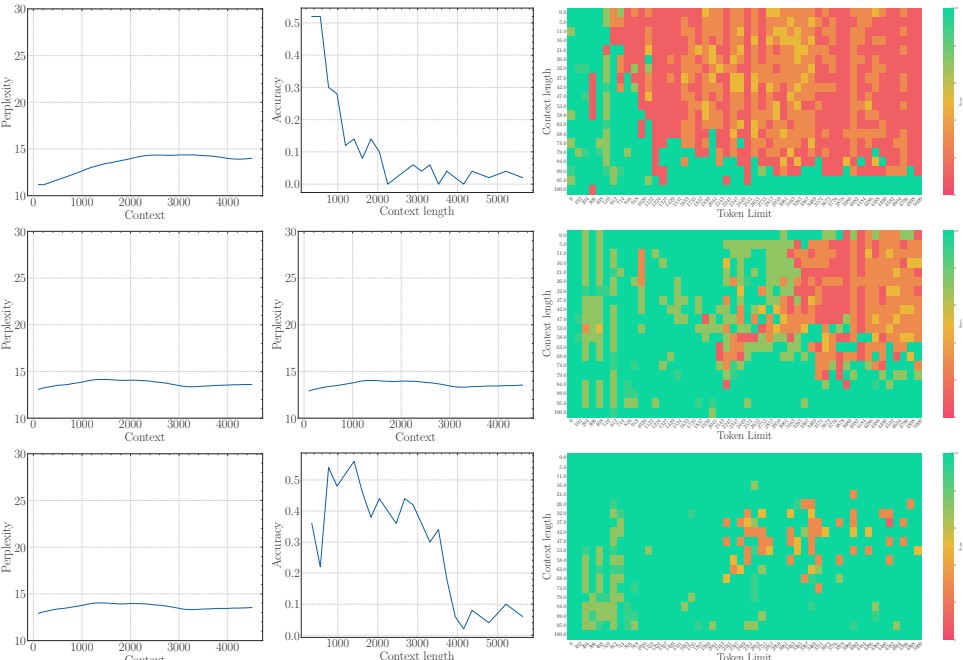

Figure 7: The first row: the results of a 2B model training from scratch with base=1e2. The second row: The results of fine-tuning the 2B model with base=1e4. The third row: The results of fine-tuning the 2B model with base=1e6.

than 32k. Further, when we increase the base to 1e6 and fine-tune the base 2B model on 32K (the third row in Figure 7), the model could obtain a larger context length than base=1e4, which is in accordance with our theory.

To further remove the influence of model size, we also fine-tuned a larger 7B model on a 32k context length with a RoPE base set to 1e4 and observed an effective context length nearly identical to that of the 2B model with the same RoPE base (see Appendix D). This is empirical proof that the effective context length is determined by RoPE's base.

### 5.4 Interpretation for the superficial long context capability for small base

Based on our theory and empirical observations, it is easy to explain what happens in Figure 3.

**Better Extrapolation (Perplexity)?** Due to the small base, $B_{m,\theta}$ can be smaller than zero as $m$ increases, which is shown in Figure 5. The model can't attend more to similar tokens than random tokens with a large relative distance, so the model tends to focus more on nearby tokens, this will lead to a smaller empirical receptive field, even smaller than the training length. In this case, the model has a strong ability to maintain perplexity stability [27].

**Worse Ability (Long-eval and NIH)!** According to our previous analysis, RoPE's base bounds the context length, and the context length bounded by 500 is much lower than that bound by 10,000. Therefore, when the base is set to 500, the effective context length drops sharply, even after training on 32k context length.

### 5.5 OOD theory is insufficient to reveal long context capability

Section 3 mentions that methods based on the OOD theory of rotation angles may not fully reflect the long context capability. In this section, we conduct further experiments to substantiate and explain this observation. We present two methods to extend the context length of Llama2 from 4k to 32k. Both of them are devoid of OOD angles. These methods are delineated mathematically as follows:

- Method 1: $\theta_i = (5e6)^{-2i/d}$,

Table 3: The comparison of "Method 1" and "Method 2". These methods are designed carefully. They both are no OOD, but they are very different under our theory.

| Method | OOD | Long-eval | | numbers of $m$ whose $B_{m,\theta} \leq 0$ | |
| --- | --- | --- | --- | --- | --- |
| | | 15k | 30k | 15k | 30k |
| Method 1 | ✘ | 0.33 | 0.27 | 0 | 0 |
| Method 2 | ✘ | 0.40 | 0.00 | 97 | 2554 |

- Method 2: $\theta_i = \begin{cases} (1e4)^{-2i/128}/8, & i \geq 44 \\ (1e4 * 8^{128/88})^{-2i/128}, & i < 44. \end{cases}$

We can see from Table 3 that these two methods exhibit significantly different long context capabilities. Under the perspective of OOD rotation angle, both methods avoid OOD rotation angle, suggesting effective extrapolation. However, despite being trained on a context length of 32k, "method 2" struggles in completing the retrieval task at a context length of 32k. This phenomenon is beyond the scope which the OOD theory can explain.

Under our perspective, "method 2" is severely violating $B_{m,\theta} \geq 0$ when $m \in [15k, 30k]$, thereby impeding its ability to achieve long-context discrimination. We speculate that the model may achieve better extrapolation in the fine-tuning stage if the base is sufficiently large to surpass a lower bound and avoid OOD of rotation angles.

# 6 Related Work

**Position embedding.** Since its introduction, Transformer [4] has achieved remarkable results in the field of natural language processing. To make full use of the order of sequence, researchers have introduced position embedding. The earliest position embedding was based on sinusoidal functions [4] for absolute positions, learnable absolute position embedding [28] and many variants [29, 30] were proposed. Nevertheless, absolute position embedding has difficulties in extending directly to texts longer than the training length. Subsequently, researchers proposed relative position embedding methods [31, 32]. With the development of large language models, rotary position embedding and its variants [6, 23] has become widely used, such as Llama2 [7], Baichuan2 [8], Mistral-7B-[33]. A recent study reveals that no position embedding is also potential [34].

**Long context learning.** Implementing models with longer or even infinitely long contexts has always been an important goal in the field of natural language processing. Due to the squared complexity of the transformer model over time, a significant portion of the work focuses on improving the model structure [35, 35, 36, 37]. However, most of the work is still based on the transformer architecture. The other part of the work is aimed at reducing the computational complexity of attention itself, such as sparse attention [38] and group query attention [39]. In addition, there are also some optimizations in engineering efficiency, such as flash attention [40] and ring attention [41]. In the model inference stage, to save time and space, there are also some methods for accelerating long context, such as KV cache compression [42], etc. And the position embedding is important in extrapolation. In the process of fine-tuning, methods such as PI [12], NTK, and YARN [11] are used to change the original position embedding information. FoT [43] assigns the position information of the tokens outside the local context as the first token in the local context.

# 7 Limitation

In this work, we investigate the relationship between the base of RoPE and context length. Although we have derived that there exists a lower bound for the base of RoPE determined by context length, the existence of the upper bound for RoPE's base remains an open question that warrants further exploration. In addition, because of the lack of effective benchmarks for assessing long-context capabilities, the scope of long-context capabilities discussed in this paper may be limited.

# 8   Conclusion

Our work presents a comprehensive study on the role of RoPE in LLMs for effectively modeling long context. Our main contribution lies in uncovering a novel property of RoPE through theoretical analysis, demonstrating that as the relative distance between tokens increases, the model's ability to attend more to similar tokens decreases. According to our theory, we derive a lower bound for RoPE's base in accommodating to expected context lengths. Our experimental results validate that the base of RoPE bounds context length for not only fine-tuning but also the pre-training stage. Our theory offers a new perspective on understanding the functionality of RoPE in long-context modeling. By shedding light on the relationship between context length and position embedding, we hope our work could provide insights for enhancing the long context capability of LLMs.

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

## A The proof of Theorem 1

Assuming that the components of query $q \in R^d$ and key $k \in R^d$ are independent, their standard deviations are denoted as $\sigma \in R^d$ and the means are donated as $\mu \in R^d$. The key $k^*$ similar to $q$ is $q + \epsilon$, where $\epsilon$ is a random variable with a mean of 0. Then, we have:

$$
\begin{aligned}
&\mathbb{E}_{q,k^*} q^T R_m k^* - \mathbb{E}_{q,k} q^T R_m k \\
=&\mathbb{E}_q q^T R_m q + \mathbb{E}_{q,\epsilon} q^T R_m \epsilon - \mathbb{E}_{q,k} q^T R_m k \\
=&\mathbb{E}_q \sum_{i=0}^{d/2-1} (q_{2i}^2 \cos(m\theta_i) - q_{2i}q_{2i+1} sin(m\theta_i) + q_{2i+1}q_{2i} sin(m\theta_i) + q_{2i+1}^2 \cos(m\theta_i)) + \mathbb{E}_q q^T R_m \mathbb{E}_\epsilon \epsilon \\
&- \mathbb{E}_{q,k} \sum_{i=0}^{d/2-1} (q_{2i}k_{2i} \cos(m\theta_i) - q_{2i}k_{2i+1} sin(m\theta_i) + q_{2i+1}k_{2i} sin(m\theta_i) + q_{2i+1}k_{2i+1} \cos(m\theta_i)) \\
=& \sum_{i=0}^{d/2-1} \mathbb{E}(q_{2i}^2) \cos(m\theta_i) - \mu_{2i}\mu_{2i+1} sin(m\theta_i) + \mu_{2i}\mu_{2i+1} sin(m\theta_i) + \mathbb{E}(q_{2i+1}^2) \cos(m\theta_i)) + \mu R_m 0 \\
&- \sum_{i=0}^{d/2-1} (\mu_{2i}^2 \cos(m\theta_i) - \mu_{2i}\mu_{2i+1} sin(m\theta_i) + \mu_i \mu_{2i+1} sin(m\theta_i) + \mu_{2i+1}^2 \cos(m\theta_i)) \\
=& \sum_{i=0}^{d/2-1} (E(q_{2i}^2 + q_{2i+1}^2) - \mu_{2i}^2 - \mu_{2i+1}^2) \cos(m\theta_i) \\
=& \sum_{i=0}^{d/2-1} (\sigma_i^2 + \sigma_{i+1}^2) \cos(m\theta_i) \quad (12)
\end{aligned}
$$

Then we can get:

$$
\sum_{i=0}^{d/2-1} (\sigma_{2i}^2 + \sigma_{2i+1}^2) \cos(m\theta_i) = \mathbb{E}_{q,k^*} q^T R_m k^* - \mathbb{E}_{q,k} q^T R_m k \quad (13)
$$

And when all $\sigma$ are equal, we can get:

$$
\sum_{i=0}^{d/2-1} \cos(m\theta_i) = \frac{1}{2\sigma^2} (\mathbb{E}_{q,k^*} q^T R_m k^* - \mathbb{E}_{q,k} q^T R_m k) \quad (14)
$$

## B The detail setting of experiment

For training, we mainly conducted experiments on Llama2-7B [7] and Baichuan2-7B [8]. In addition, we also trained a 2B model from scratch, whose structure is the same as Baichuan2-7B-Base but with a smaller hidden size = 2048. Both training and testing are accelerated by FlashAttention-2 [40] and Megatron-LM [44]. The dataset of both fine-tuning and training from scratch is a subset of RedPajama [24]. The hyperparameters of training are listed in Appendix 4. All experiments are conducted on a cluster of 16 machines with 128 NVIDIA A100 80G.

Table 4: Training hyper-parameters in our experiments

| Model | Training length | Training tokens | Batchsize | Base LR | LR decay | Weight decay |
|---|---|---|---|---|---|---|
| Llama2-7B-Base | 32K | 4B | 128 | 2e5 | constant | 0 |
| Baichuan2-7B-Base | 32K | 4B | 128 | 2e5 | constant | 0 |
| Our-2B-Base | 4K | 1T | 1024 | 2e4 | cosine | 0.1 |

**Question**: Below is a record of lines I want you to remember. Each line begins with 'line <line index>' and contains a '<REGISTER_CONTENT>' at the end of the line as a numerical value. For each line index, memorize its corresponding <REGISTER_CONTENT>. At the end of the record, I will ask you to retrieve the corresponding <REGISTER_CONTENT> of a certain line index. Now the record start:

...

line swift-baby: REGISTER_CONTENT is <12821>
line dangerous-breast: REGISTER_CONTENT is <28051>
line bad-sculptural: REGISTER_CONTENT is <32916>
line flashy-college: REGISTER_CONTENT is <34027>
line voiceless-brochure: REGISTER_CONTENT is <8964>
line fast-peony: REGISTER_CONTENT is <5218>

...

Now the record is over. Tell me what is the <REGISTER_CONTENT> in line dangerous-breast? I need the number. **Answer**:

Figure 8: Long-eval sample prompt

For evaluation, we test the long context capabilities comprehensively, the benchmarks are listed below: **perplexity** on PG19 [25] test split. We evaluate the perplexity of each sample and get the mean value across samples.

**Long-eval** [20]. This test generates massive random similar sentences and asks the model to answer questions according to a specific sentence in the context. Because the long context consists of many similar patterns, it's more difficult to get the right answer. We find this test is harder than other long context evaluations such as Perplexity, Passkey Retrieval [45], Needle in Haystack [21]. A test sample is list in Figure 8

**needle in haystack(NIH)** [21]. NIH tests the long context capability not only under different context lengths but also at different positions where the correct answer is located in the context, which provides a more detailed view of the long context capability.

## C Baichuan2-7B-Base: Lower bound Base of RoPE

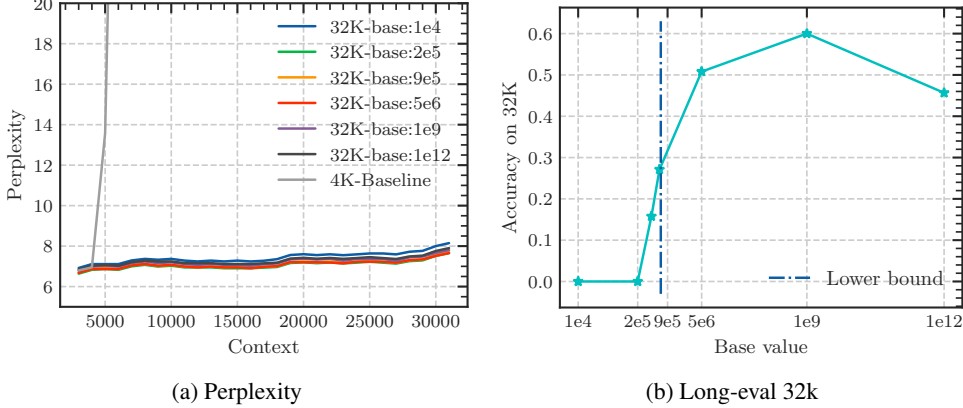

(a) Perplexity

(b) Long-eval 32k

Figure 9: Fine-tuning Baichuan2-7B-Base on 32k context length with varying RoPE's base. Although the perplexity remains low with varying bases, the Long-eval accuracy reveals a discernible bound for the base value, below which the Long-eval accuracy declines significantly. the dotted line denotes the lower bound derived from Eq. 11.

## D Long Context Test Results on Various LLMs

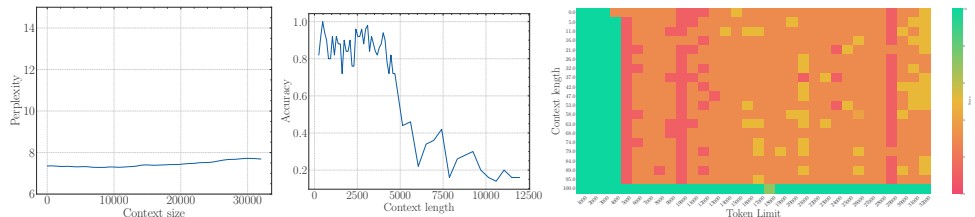

Figure 10: Llama2-7B-Base with base=1e4 fine-tuned on 32k context (original context=4096)

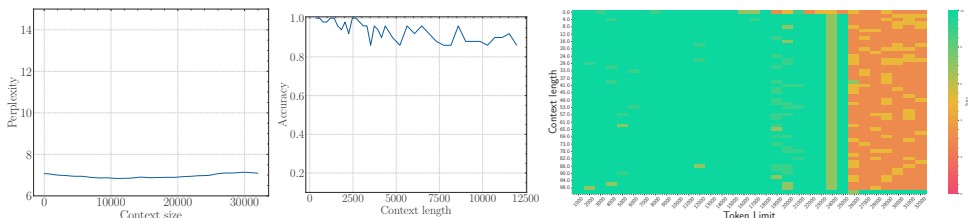

Figure 11: Llama2-7B-Base with base=2e5 fine-tuned on 32k context (original context=4096)

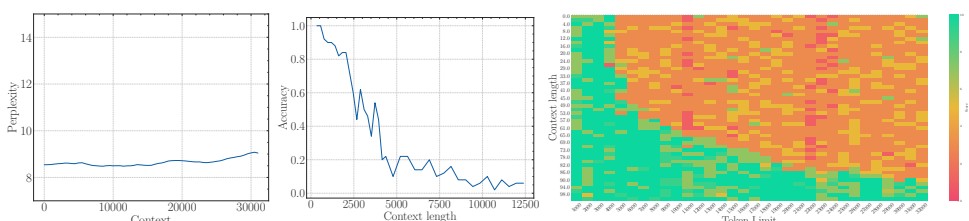

Figure 12: Baichuan2-7B-Base with base=1e4 fine-tuned on 32k context (original context=4096)

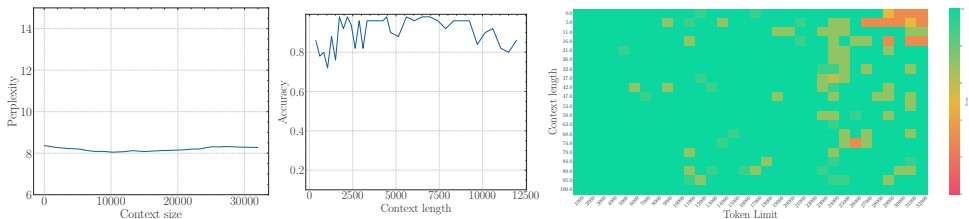

Figure 13: Baichuan2-7B-Base with base=2e5 fine-tuned on 32k context (original context=4096)

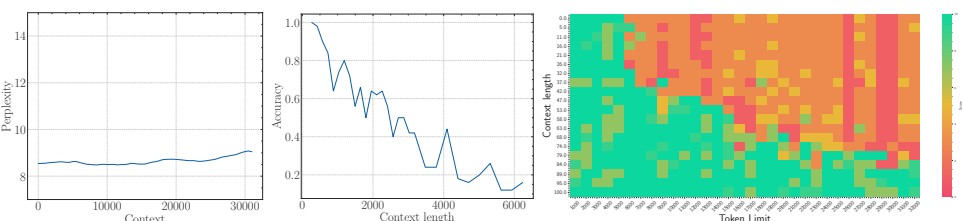

Figure 14: Qwen1.5-7B-Base [9] with base=1e4 fine-tuned on 32k context (original context=4096)

# E The python code for calculating the low bound base for a context length of 32k

```python
"""the python code for calculate the
low bound base for a context length of 32k"""
import torch
import numpy as np
def get_BMtheta_expectation(base,context_size=2**15,dim=128):
    realdim = dim / 2
    d= torch.arange(0, realdim, 1)
    theta = base ** (-2*d/dim)
    dist= torch.outer(torch.arange(0,context_size),theta).cos()
    return dist.sum(dim=1) / realdim
search_base = []
for x in range(3,10):
    for i in range(1,10):
        for j in range(10):
            search_base.append((i+j/10)* (10**x))
for base in search_base:
    ans = get_BMtheta_expectation(base)
    if True not in (ans<0):
        print("Find!Base=", base)
        break
    idx = np.argmax(ans < 0)
    print('base', base, 'first zero position', idx)
```

# F A empirical verification of Desiderata 2

The **Desiderata 2** introduced in Section 4 is intuitively plausible, but its empirical validity requires verification. To investigate this, we conducted a detailed empirical analysis. The similarity between tokens is measured by the cosine similarity (denoted as A) of their corresponding hidden states, while the attention allocation between tokens is governed by the attention score (denoted as B). The desiderata "similar tokens receive more attention" implies that a higher value of **A** should lead to a higher value of **B**.

To test this desiderata, we performed experiments using Llama1-7B, Llama2-7B, and Llama3-8B models. We selected 200 segments from the PG19 dataset, each containing 1024 tokens, and computed Spearmans rank correlation coefficient between (A) and (B). A positive correlation coefficient would indicate that as token similarity (A) increases, the corresponding attention score (B) also increases. The magnitude of the coefficient reflects the strength of this correlation.

The results, presented in Figure 15 confirm that Spearmans rank correlation coefficient is positive, validating the desiderata that "similar tokens receive more attention". Furthermore, we observe that this positive correlation is more pronounced in the Llama3-8B model compared to Llama2-7B and Llama1-7B, suggesting that larger and more advanced models are better at capturing this relationship.

# G Evaluation results on RULER

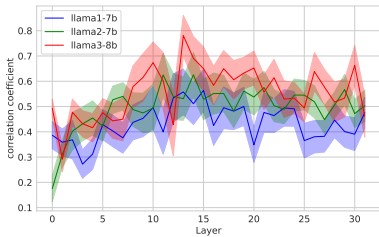

Figure 15: Spearmans rank correlation coefficient between the similarity and the attention score of two tokens. The thick line represents the mean $\mu$ calculated from different samples. The upper and lower boundaries of the line are $\mu + \sigma$ and $\mu - \sigma$, respectively, where $\sigma$ is the standard deviation of different samples.

Table 5: Evaluation results on RULER. We finetune Llama2-7b to 32k context length (the low bound base is 6e5) using different RoPE's bases. NS is short for NIAH-single and NM is short for NIAH-Multikey.

| Base | Context Len. | Sub tasks | | | | | | | | | | | | | Ave. |
| | | NS-1 | NS-2 | NS-3 | NM-1 | NM-2 | NM-3 | NIAH_Multivalue | NIAH_Multiquery | VT | CWE | FWE | QA1 | QA2 | |
|---|---|---|---|---|---|---|---|---|---|---|---|---|---|---|---|
| 500 | 4k | 23.0 | 31.0 | 26.0 | 31.0 | 13.0 | 11.0 | 73.0 | 75.0 | 2.0 | 59.5 | 53.7 | 51.0 | 29.0 | 36.78 |
| | 8k. | 15.0 | 18.0 | 11.0 | 14.0 | 2.0 | 3.0 | 50.0 | 37.0 | 1.0 | 46.3 | 46.0 | 18.0 | 21.0 | 21.72 |
| | 16k | 5.0 | 8.0 | 5.0 | 9.0 | 2.0 | 1.0 | 28.0 | 33.0 | 0.0 | 23.0 | 40.7 | 23.0 | 27.0 | 15.74 |
| | 32k | 1.0 | 1.0 | 2.0 | 4.0 | 1.0 | 1.0 | 10.0 | 12.0 | 0.0 | 1.5 | 22.7 | 16.0 | 24.0 | 7.40 |
| 1e4 | 4k | 99.0 | 100 | 96.0 | 91.0 | 85.0 | 65.0 | 66.0 | 99.0 | 90.0 | 34.1 | 77.33 | 66.0 | 44.0 | 77.88 |
| | 8k. | 53.0 | 55.0 | 58.0 | 59.0 | 34.0 | 4.0 | 49.0 | 84.0 | 1.0 | 33.7 | 27.67 | 30.0 | 29.0 | 39.80 |
| | 16k | 21.0 | 24.0 | 28.0 | 36.0 | 17.0 | 3.0 | 72.0 | 75.0 | 0.0 | 49.3 | 8.67 | 10.0 | 25.0 | 28.38 |
| | 32k | 5.0 | 8.0 | 11.0 | 13.0 | 7.0 | 0.0 | 38.0 | 39.0 | 0.0 | 17.1 | 1.33 | 19.0 | 26.0 | 14.19 |
| 2e5 | 4k | 100 | 100 | 100 | 97.0 | 97.0 | 77.0 | 99.0 | 99.0 | 100 | 79.6 | 86.0 | 45.0 | 45.0 | 86.51 |
| | 8k. | 100 | 100 | 100 | 100 | 96.0 | 48.0 | 97.0 | 100 | 100 | 42.9 | 65.00 | 44.0 | 40.0 | 79.46 |
| | 16k | 100 | 100 | 100 | 97.0 | 74.0 | 23.0 | 92.0 | 100 | 97.0 | 20.7 | 8.33 | 38.0 | 37.0 | 68.23 |
| | 32k | 99.0 | 100.0 | 95.0 | 95.0 | 32.0 | 9.0 | 62.0 | 87.0 | 82.0 | 27.0 | 39.0 | 29.0 | 38.0 | 61.08 |
| 6e5 | 4k | 100 | 100 | 100 | 97.0 | 96.0 | 65.0 | 99.0 | 100 | 100 | 84.6 | 90.0 | 52.0 | 49.0 | 87.12 |
| | 8k. | 100 | 100 | 100 | 99.0 | 96.0 | 40.0 | 93.0 | 100 | 100 | 43.4 | 66.33 | 34.0 | 47.0 | 78.36 |
| | 16k | 100 | 100 | 100 | 95.0 | 74.0 | 37.0 | 93.0 | 99.0 | 98.0 | 27.4 | 62.67 | 37.0 | 41.0 | 74.16 |
| | 32k | 100 | 100 | 94.0 | 96.0 | 47.0 | 12.0 | 70.0 | 89.0 | 97.0 | 20.5 | 63.67 | 25.0 | 39.0 | 65.63 |
| 9e5 | 4k | 100 | 100 | 99.7 | 97.0 | 95.1 | 71.0 | 99.0 | 99.7 | 100 | 83.6 | 88.5 | 49.3 | 46.9 | 86.91 |
| | 8k. | 100 | 100 | 100 | 98.4 | 96.3 | 48.4 | 92.7 | 100 | 100 | 44.66 | 67.53 | 35.8 | 46.7 | 79.27 |
| | 16k | 100 | 100 | 100 | 93.8 | 78.8 | 42.4 | 90.9 | 99.3 | 98.6 | 27.58 | 59.97 | 36.4 | 40.1 | 74.45 |
| | 32k | 100 | 100 | 95.8 | 96.3 | 52.7 | 18.3 | 64.3 | 89.6 | 97.9 | 17.26 | 63.77 | 26.2 | 39.0 | 66.24 |
| 5e6 | 4k | 100 | 100 | 99.0 | 97.0 | 93.0 | 85.0 | 99.0 | 99.0 | 100.0 | 81.2 | 85.0 | 43.0 | 42.0 | 86.40 |
| | 8k | 100 | 100 | 100 | 97.0 | 97.0 | 68.0 | 92.0 | 100 | 100 | 47.6 | 70.3 | 40.0 | 46.0 | 81.38 |
| | 16k | 100 | 100 | 100 | 100 | 91.0 | 90.0 | 55.0 | 86.0 | 100 | 100 | 28.0 | 53.7 | 35.0 | 79.90 |
| | 32k | 100 | 100 | 100 | 97.0 | 66.0 | 33.0 | 51.0 | 91.0 | 100.0 | 9.7 | 64.0 | 29.0 | 39.0 | 67.67 |
| 1e9 | 4k | 100 | 100 | 100 | 95.0 | 96.0 | 72.0 | 100 | 99.0 | 67.0 | 63.8 | 77.7 | 41.9 | 29.0 | 80.11 |
| | 8k | 100 | 100 | 100 | 96.0 | 90.0 | 54.0 | 95.0 | 100 | 88.0 | 35.0 | 60.0 | 28.0 | 35.0 | 75.46 |
| | 16k | 100 | 100 | 100 | 96.0 | 77.0 | 43.0 | 83.0 | 100 | 72.0 | 23.7 | 51.3 | 27.0 | 35.0 | 69.85 |
| | 32k | 100 | 100 | 100 | 93.0 | 69.0 | 23.0 | 58.0 | 92.0 | 94.0 | 18.1 | 55.7 | 17.0 | 35.0 | 65.75 |

