# OpenReview forum: "Base of RoPE Bounds Context Length"
_NeurIPS.cc/2024/Conference — NeurIPS 2024 poster_

### Official Review · Reviewer_aPSb · 2024-07-07

**Soundness:** 4
**Presentation:** 3
**Contribution:** 3
**Rating:** 7
**Confidence:** 5

**Summary:**

The paper investigates the role of RoPE in long-context LLMs. It highlights that the base of RoPE crucially affects the model's ability to handle long contexts. This paper derives a theoretical (and empirical) lower bound for the base value required to maintain long-context capabilities and validates this through empirical experiments with models like Llama2-7B and Baichuan2-7B, and a 2B model trained from scratch. This work offers insights into the base of RoPE for long-context processing in LLMs, which is inspiring for the development and design of long-context LLMs.

**Strengths:**

1. The lack of long-context ability is still an under-explored question and a very important one. Thus, the research topic of this paper is both theoretically important and practical.
2. The relationship between the RoPE base and long-context ability is important, and the proposed lower bound is interesting and inspiring.
3. The experiments in this paper are comprehensive and well support the claims made in the paper.
4. The presentation of this paper is overall very good and easy to understand and follow.
5. The claims of this paper are inspiring for the development of long-context LLM.
6. I am also very interested in RoPE-based selection in LLM and like this paper.

**Weaknesses:**

1. The Desiderata 2. The similar token gets more attention in Section 4 seems intuitively correct but may not be empirically correct. As this desideratum is the fundamental motivation of Theorem 1, a thorough empirical verification is a must-have.
2. The motivation section is not well-written or organized. It confuses me while I read this part. These are just some previous empirical observations and are not deeply discussed. As the motivation part is very important, I would like to see a clear and well-organized motivation.
3. For the Caption of Figure 6, it would be better to show how to derive the value of the dotted line. I would also like to see a detailed derivation here.
4. Will the author plan to release the models (including the fine-tuned Llama2, Baichun2 with varying lengths, and the 2b model series)? This would benefit future work.
5. I wonder about the negativeness and positiveness of the $q^{T}R_{m,\theta}k$ in equations (8) and (9). If the values are negative, say -2 and -1, which one indicates more attention?
6. For section 5.3, I would like to regard this as a conjecture rather than an explanation.
7. Moinries
    - The upper case and lower case of the title are not consistent, such as the title of Section 4.1 and Section 2.2
    - Line 225 “method2” -> “Method 2”

**Questions:**

See **Weaknesses**

**Limitations:**

See **Weaknesses**

---

> ### Author Rebuttal · Authors · 2024-08-05
>
> **Q1:The Desiderata 2. The similar token gets more attention in Section 4 seems intuitively correct but may not be empirically correct. A thorough empirical verification is a must-have.**
>
> A1: In our paper, the similarity between tokens measured by the cosine similarity(A) of their corresponding hidden-states,  and the amount of attention is determined by the attention score(B). "Similar token gets more attention" means that a larger (A) leads to a larger (B).
>
> To validate this desiderata, we conducted experiments on Llama1-7B, Llama2-7B, and Llama3-8B. We extracted 200 segments from PG19, each consisting of 1024 tokens, and calculated the Spearman’s rank correlation coefficient between (A) and (B). A positive Spearman’s rank correlation coefficient  indicates that as (A) increases, so does (B), and the higher the absolute value of the coefficient, the stronger the positive correlation.
>
> The results are shown in  Figure 1 in PDF of Global Response.  The positive  Spearman’s rank correlation coefficient  validates the property of "similar token gets more attention". And we observe that this positive correlation is stronger in Llama3-8B compared to Llama2-7B and Llama1-7B, suggesting that more powerful models may better learn this positive correlation.
>
> **Q2:The motivation section is not well-written or organized. It confuses me while I read this part. These are just some previous empirical observations and are not deeply discussed.**
>
> A2: Thank you for your suggestions. We will reorganize the motivation in the revised version as follows:
>
> Recent advancements in long-context language models have seen widespread adoption of NTK-based methods [6, 12, 13]. However, a curious trend has emerged: practitioners often employ significantly larger base values than those originally suggested by NTK-aware approaches. This discrepancy raises critical questions about the efficacy of current theoretical frameworks. Why do practitioners deviate from the recommendations of NTK-based methods? Is the out-of-distribution (OOD) theory underlying these methods insufficient to fully unlock long-context capabilities?
>
> On the other hand, recent research [14], driven by OOD theory, proposes using a much smaller base for RoPE to extend context length. However, our findings, as illustrated in Figure 3, suggest that this approach may only provide superficial long-context capability[21]. While achieving low perplexity even at 128k context length (explicable by OOD theory), the model fails to retrieve relevant information for context lengths as short as 1k—well below its pre-trained length. The observation suggests that the small base determined by OOD theory can't unlock true long-context capability.
>
> These phenomena motivate us to delve deeper into the relationship between RoPE's base and context length. To address the gap between OOD theory and our observations, we conduct a theoretical exploration in the next section, aiming to uncover the underlying mechanisms of effective long-context modeling.
>
> **Q3: For the Caption of Figure 6, it would be better to show how to derive the value of the dotted line.**
>
> A3: Thank you for pointing out this. We apologize for not providing specific procedures and pseudocode for calculating the values in the paper. We will provide detailed information in the revised version.
>
> In our paper, we adopted an imprecision search. Specifically, we traversed the base values of $a.b\times 10^x$, where a, b, and x are all Arabic numerals, e.g. $2.3\times 10^6$. We then observed whether $B_{\theta, m} $ can always be non negative within the window length and select the minimum base that meets the conditions. The python code is in the in PDF of Global Response.
>
> **Q4:Will the author plan to release the models? This would benefit future work.**
>
> A4: We are happy to open source these models and hope that they can benefit future work. However, due to the requirements of rebuttal, we are unable to provide a download link here. After the decision is made on the paper, we will release all the models used in the paper.
>
> **Q5: I wonder about the negativeness and positiveness of the in equations (8) and (9). If the values are negative, say -2 and -1, which one indicates more attention?**
>
> A5: In equations (8) and (9), -1 indicates more attention.
>
> **Q6: For section 5.3, I would like to regard this as a conjecture rather than an explanation.**
>
> A6: It seems that we didn't provide much explanation in section 5.3. In section 5.3, we train a 2B model with small base from scratch and find that it has a pool long context capability, which is consistent with our theoretical perspective.
>
> Perhaps you are referring to section 5.4, so we answer according to the content of section 5.4 below.
>
> Thank you for pointing it out. We agree that this is not a strict explanation. There are many different perspectives on positional encoding, and we are providing an interpretation for the strange phenomenon "superficial long context capability for small base" from the theory perspective proposed in our paper. This isn't a strict proof, and further research is needed on the performance of models and positional encoding.
>
> **Q7:Moinries: The upper case and lower case of the title are not consistent, such as the title of Section 4.1 and Section 2.2; Line 225 "method2" -> "Method 2"**
>
> A7: Thank you for carefully reviewing our paper. We will address the formatting and grammatical errors in revisions.

---

### Official Review · Reviewer_XTUy · 2024-07-09

**Soundness:** 4
**Presentation:** 4
**Contribution:** 4
**Rating:** 8
**Confidence:** 4

**Summary:**

ROPE is wildily employed in popular LLMs which encodes positional information with a rotation matrix. Although RoPE is used to enhance long-context capabilities by adjusting its base parameter to address OOD issues, this paper finds that this may result in only superficial long-context abilities. Authors re-evaluate RoPE's role and introduce a novel property of long-term decay, showing that the base of RoPE limits context length, with an absolute lower bound required for certain capabilities. This work clarifies the theoretical and empirical relationship between context length and the RoPE base, offering insights for future long-context training.

**Strengths:**

- The two provided desideratas are highly logical and well-aligned with language modeling. The assumption made in this paper is quite reasonable and closely aligned with practical scenarios.

- Insightful analysis. The theoretical results presented in this paper are easy to understand. To the best of my knowledge, the final bound for the ROPE base is novel and first introduced in this paper

**Weaknesses:**

There is no obvious weaknesses in my opinion. I just have a few questions:

- Regarding the "Desiderata 2 The similar token gets more attention", recently StreamLLM [1] shows that there exists "attention sink" in popular LLMs. Namely most of tokens attend to the first few tokens. This somehow contradicts with the principle that "the similar token gets more attention". Could you provide your thoughts on this statement?

- Anthropic's blogs reveal that different heads may have different functionality in in-context learning. How may this interplay with the Rope base? Do you think different heads may have different optimal rope base?

[1] Efficient Streaming Language Models with Attention Sinks
[2] In-context Learning and Induction Heads

**Questions:**

See Weaknesses

**Limitations:**

See Weaknesses

---

> ### Author Rebuttal · Authors · 2024-08-05
>
> **Q1:Regarding the "Desiderata 2 The similar token gets more attention", recently StreamLLM  shows that there exists "attention sink" in popular LLMs. Namely most of tokens attend to the first few tokens. This somehow contradicts with the principle that "the similar token gets more attention". Could you provide your thoughts on this statement?**
>
> A1:
> Firstly, we think that they are not in conflict in phenomenon. We think that the sink token can be considered as a form of regularization to focus on and serves as an anchor point for positional information.  “most of tokens attend to the first few tokens” and  "the similar token gets more attention" are two distinct properties refer to different aspects. The former means "current token tends to pay more attention to the first few tokens than the tokens in other positions",  the latter means "At the same relative position, current token tends to pay more attention to the similar token than random tokens". They can actually both be satisfied at the same time. For example, when similar token exists, the current token give a weight of 0.8 to first token and 0.2 to similar token. When similar token does not exist, the current token give a weight of 1.0 to first token and 0 to other tokens.
>
> More importantly, as StreamLLM say "While StreamingLLM improves the efficiency of LLMs in streaming contexts, it does not extend the models’ context window or enhance their long-term memory capabilities. "  "most of tokens attend to the first few tokens" may lead to low perplexity, but can't extend the context window.  While based on "the similar token gets more attention than random ones", we can adjust RoPE's base to achieve good performance on more challenge benchmarks rather than only low perplexity.
>
> **Q2: Anthropic's blogs reveal that different heads may have different functionality in in-context learning. How may this interplay with the Rope base? Do you think different heads may have different optimal rope base?**
>
> A1:  According to our research, for example, in the induction heads mechanisms proposed in Anthropic's blogs,  the head that retrieves long-distance information may require a larger base, while a copy head that simply copies the information of the previous token and only focuses on nearby tokens may only require a small base. So we believe that different heads require different optimal base. However, setting different bases for  different heads is a highly challenging task, perhaps a search method similar to LongRoPE[1] can be used.
>
> [1] LongRoPE: Extending LLM Context Window Beyond 2 Million Tokens

---

> > ### Comment · Reviewer_XTUy · 2024-08-09
> >
> > Thank you for your clarification. I am satisfied with the response.
> >
> > Good Luck!

---

### Official Review · Reviewer_3a2j · 2024-07-13

**Soundness:** 3
**Presentation:** 3
**Contribution:** 3
**Rating:** 6
**Confidence:** 3

**Summary:**

This paper investigates the role of Rotary Position Embedding (RoPE) in Large Language Models (LLMs), with a focus on the relationship between RoPE's base and the model's long context ability.
The study looks into the long-context abilities and limitations of current methods that rely on smaller RoPE bases. With a lower base, LLM models may exhibit superficial long-context abilities, achieving low perplexity but failing to retrieve relevant information in extended contexts.
Theoretically, The paper establishes two desiderata for the attention mechanism in language modeling: 1.Closer tokens receive more attention. 2.Similar tokens receive more attention. It examines these when applying RoPE to LLMs, revealing a long-term decay in attention score and the ability to differentiate similar from random tokens. This leads to a theorem indicating that RoPE’s base sets an absolute lower bound for achieving specific context lengths.

**Strengths:**

1. Clarity and Technical Correctness: The paper is clear and technically sound, with theoretical and empirical analyses.

2. Experimental Rigor and Reproducibility: It includes extensive experiments to back up the theoretical findings, along with detailed setup and results.

3. Novel Findings: This paper presents a critical study of whether we should use smaller bases for continuous training, as suggested by previous work. Furthermore, this paper presents a novel perspective on long-term decay, as well as an absolute lower bound on the RoPE base parameter required for specific context lengths. This adds new knowledge to the field and improves our understanding of position embedding in LLMs.

**Weaknesses:**

Overall, I like this paper. I would like to suggest that the authors strengthen their work by considering the following points:

1. Extensively test the model using benchmarks such as RULER [1].

2. Provide more empirical observations on the relationship between the base of RoPE and model performance on those challenging benchmarks.


--

[1] https://github.com/hsiehjackson/RULER

**Questions:**

For the suggestions, please refer to the weaknesses section.

---

> ### Author Rebuttal · Authors · 2024-08-05
>
> **Q:Extensively test the model using benchmarks such as RULER. Provide more empirical observations on the relationship between the base of RoPE and model performance on those challenging benchmarks.**
>
> A: We greatly appreciate your suggestion. We evaluated  Llama2-7b on RULER, and the evaluation results are shown in Table 1. We can observe that when the base value exceeds the lower bound 6e5 given in our paper for 32k context, the model can achieve better performance on the RULER. And when base is greater than 6e5, the improvement is slight. The detailed evaluation results on various sub tasks are presented in Table 2.
> We will include these results in the revisions of our paper.
>
> **Table1. Comparison on Ruler, when fine-tuning Llama2-7b to a length of 32k ( the low bound base is 6e5) under different settings of RoPE's base**
> | Base || Length |  | |
> | - | - | - | - | - |
> | |4k|8k|16k|32k|
> | 500   | 36.78   | 21.72|15.74|7.40|
> | 1e4  | 77.88   | 39.80  | 28.38  | 14.19   |
> | 2e5   | 86.51 | 79.45  | 68.23  |  61.08 |
> | 6e5   | **87.12**   | 78.36  | 74.16  | 65.63  |
> | 9e5   | 86.76   | **79.87**  | **74.64**  | **66.65**  |
> | 5e6   | 86.91 | 79.27 | 74.45  | 66.24|
> | 1e9   | 80.04  | 75.46 | 69.85  |65.75 |
>
>
> **Table2. The detail results on sub-tasks of RULER. Niah_single is short for ns. Niah_multikey is short for nm.**
> |base|||ns1|ns2|ns3|nm1|nm2|nm3|niah_multivalue|niah_multiquery|vt|cwe|fwe|qa_1|qa_2
> | - | - | - | - | - |  - | - | - | - | - | - | - | - | - | - |  - |
> |500| length|Average Score|  |  |   |  |  |  |  |  |  |  |  |  |   |
> | | 2k |46.04| 57.0 |67.0| 48.0  |45.0  | 17.0 | 8.0 | 83.0 | 89.0 | 6.0 | 44.5 |45.0  |57.0  | 32.0  |
> | | 3k |34.04| 36.0 | 31.0 | 19.0  | 31.0 | 13.0 |3.0  | 74.0 | 67.0 | 3.0 |20.8  | 57.7 | 57.0 |  30.0 |
> | | 4k |36.78| 23.0 | 31.0 | 26.0  |31.0  | 13.0 | 11.0 | 73.0 | 75.0 | 2.0 | 59.5 | 53.7 | 51.0 |  29.0 |
> | | 8k | 21.72|15.0|18.0|11.0|14.0|2.0|3.0|50.0|37.0|1.0|46.3|46.0|18.0|21.0|
> | |16k|15.74|5.0|8.0|5.0|9.0|2.0|1.0|28.0|33.0|0.0|23.0|40.7|23.0|27.0|
> | |32k|7.40|1.0|1.0|2.0|4.0|1.0|1.0|10.0|12.0|0.0|1.5|22.7|16.0|24.0|
> |1e4| length|Average Score|  |  |   |  |  |  |  |  |  |  |  |  |   |
> | | 4k |77.88| 99.0 | 100 | 96.0  | 91.0 | 85.0 | 65.0 | 66.0 | 99.0| 90.0|  34.1|77.33  | 66.0 | 44.0  |
> | | 8k |39.80| 53.0 | 55.0 | 58.0  |59.0  |34.0  | 4.0 | 49.0 | 84.0 | 1.0 | 33.7 |27.67  | 30.0 | 29.0  |
> | | 16k |28.38| 21.0 | 24.0 | 28.0 | 36.0 | 17.0|3.0 | 72.0| 75.0| 0.0| 49.3| 8.67| 10.0|25.0 |
> | | 32k |14.19|5.0 | 8.0| 11.0| 13.0|7.0|0.0|38.0|39.0|0.0| 17.1|1.33|19.0|26.0|
> |2e5| length|Average Score|  |  |   |  |  |  |  |  |  |  |  |  |   |
> | | 4k |86.51| 100 | 100 | 100  | 97.0 |97.0  | 77.0 | 99.0 |99.0  |100  |79.6  | 86.0 | 45.0 | 45.0  |
> | | 8k |79.46| 100 |100  | 100  |100  | 96.0 | 48.0 | 97.0 | 100 | 100 | 42.9 | 65.00 |44.0  | 40.0  |
> | | 16k |68.23| 100 | 100 | 100  |97.0  | 74.0 | 23.0 | 92.0 | 100 |97.0 | 20.7 | 8.33 | 38.0 | 37.0  |
> | | 32k |61.08|99.0  |100.0  | 95.0  | 95.0 | 32.0 |9.0  | 62.0 | 87.0 | 82.0 |27.0  | 39.0 |29.0  |38.0   |
> |6e5| length|Average Score|  |  |   |  |  |  |  |  |  |  |  |  |   |
> | | 4k |87.12|100  |100  |100   |97.0  | 96.0|65.0| 99.0|100|100|84.6|90.0| 52.0|49.0 |
> | | 8k |78.36| 100| 100| 100 | 99.0 |96.0  |40.0  | 93.0 |100| 100|43.4 |66.33|34.0|47.0 |
> | | 16k |74.16|100|100| 100| 95.0| 74.0|37.0| 93.0|99.0|98.0|27.4| 62.67|37.0| 41.0 |
> | | 32k |65.63|100| 100| 94.0 |96.0 | 47.0 | 12.0 |70.0  |89.0  |97.0  | 20.5 | 63.67 | 25.0 |39.0   |
> |9e5| length|Average Score|  |  |   |  |  |  |  |  |  |  |  |  |   |
> | | 4k |86.91| 100 | 100 | 99.7  |97.0  |95.1  |71.0  |99.0  |99.7  |100  | 83.6 |88.5  |49.3  |46.9|
> | | 8k |79.27| 100 |100  |100   |98.4  |96.3  |48.4  |92.7  | 100 | 100 |44.66  |67.53  |35.8  |46.7|
> | | 16k |74.45| 100 |100  |100   |93.8  |78.8  |42.4  |90.9  |99.3  |98.6  |27.58  |59.97  |36.4  |40.1|
> | | 32k |66.24| 100|100  |95.8   |96.3  |52.7  |18.3  |64.3  |89.6  |97.9  | 17.26 | 63.77 |26.2  |39.0|
> |5e6| length|Average Score|  |  |   |  |  |  |  |  |  |  |  |  |   |
> | | 4k |86.40|100  |100  |99.0|97.0|93.0|85.0|99.0|99.0|100.0|81.2|85.0|43.0|42.0|
> | | 8k |81.38|100  |100  | 100  | 97.0|97.0|68.0|92.0|100|100|47.6|70.3|40.0|46.0|
> | | 16k |75.13|100  |100  | 100  |100|91.0|90.0|55.0|86.0|100|100|28.0|53.7|35.0|38.0|
> | | 32k |67.67| 100 |100  | 100  |97.0|66.0|33.0|51.0|91.0|100.0|9.7|64.0|29.0|39.0|
> |1e9| length|Average Score|  |  |   |  |  |  |  |  |  |  |  |  |   |
> | | 4k |80.04| 100  |100  |100   | 95.0|96.0  | 72.0|100  |99.0  |67.0  |63.8  |77.7  |41.9  |29.0   |
> | | 8k |75.46| 100  |100  | 100  | 96.0|90.0  |54.0  |95.0  |100  |88.0  |35.0  |60.0  |   28.0| 35.0|
> | | 16k |69.85| 100  |100  | 100  |96.0|77.0|43.0 |83.0  |100  | 72.0 |23.7 |51.3 | 27.0 |35.0 |
> | | 32k |65.75|  100 |100  | 100  |93.0|69.0|23.0 |58.0 |92.0|94.0| 18.1|55.7| 17.0| 35.0 |

---

> > ### Comment · Reviewer_3a2j · 2024-08-11
> >
> > Thank you for your response. Could you please clarify the setup used for fine-tuning Llama2-7b to a sequence length of 32k? Specifically, I’m interested in understanding which data was used and the detailed training configuration. The number you reported appears to be *higher* than what I observed in my previous experiments.

---

> > > ### Author Response · Authors · 2024-08-12
> > >
> > > Thank you for your feedback. We appreciate the opportunity to clarify and address your concerns.
> > >
> > > 1. **Data**: As mentioned in Section 5.1 and Appendix B, our training data is a subset of RedPajama. We upsampled long data during fine-tuning, with ~50% of tokens from documents ≥4k in length. This approach ensures a balance of shorter and longer data, differing from Longlora[1] which used only 8k to 32k data when fine-tuning LLaMA-2-7B to 32k. While data proportion isn't our paper's primary focus, we believe using a sampling method similar to Longlora would yield comparable conclusions.
> > >
> > > 2. **Training Configuration**:
> > >    - As stated in Section 5.1 and Appendix B, we used a fixed learning rate of 2e-5, global batch size of 128, and fine-tuned for 1000 steps. We've reviewed our configuration and can provide additional details:
> > >
> > >    | Parameter | Value |
> > >    |-----------|-------|
> > >    |Global Batch Size| 128 |
> > >    | Steps | 1000 |
> > >    | Tensor Parallelism (Tp) | 4 |
> > >    | Pipeline Parallelism (Pp) | 2 |
> > >    | Precision | bf16 |
> > >    | Learning Rate (Lr) | 2e-5 |
> > >    | Weight Decay (Wd) | 0 |
> > >    | Adam Beta1 | 0.9 |
> > >    | Adam Beta2 | 0.98 |
> > >    | Gradient Clip | 1.0 |
> > >
> > > Perhaps you can tell us the training configuration you have conducted before and the number you observed, and we can check together what caused the difference from your previous observations.
> > >
> > > Furthermore, as mentioned in our response to Reviewer XWTB, we plan to open-source our trained models. You can download them and compare them with your own trained models at that time.
> > >
> > > We look forward to your response and the opportunity for further discussion.
> > >
> > > [1] LongLoRA: Efficient Fine-tuning of Long-Context Large Language Models

---

> > > > ### Comment · Reviewer_3a2j · 2024-08-13
> > > >
> > > > Thank you for the detailed explanation regarding the data and configuration. I would encourage you to incorporate our discussion into the final version of your paper. Solid work overall!
> > > >
> > > > I have updated my score accordingly. Good luck!

---

### Official Review · Reviewer_mP48 · 2024-07-15

**Soundness:** 3
**Presentation:** 3
**Contribution:** 3
**Rating:** 6
**Confidence:** 1

**Summary:**

Hi Area Chair, I am not qualified for the review of this paper, this is out of my knowledge scope.

**Strengths:**

n/a

**Weaknesses:**

n/a

**Questions:**

n/a

---

### Author Rebuttal · Authors · 2024-08-05

Dear Reviewers, Area Chairs, and Program Chairs:

We would like to express our gratitude to all reviewers for taking their valuable time to review our paper. We sincerely appreciate all reviewers for their positive comments on our theoretical analysis, technique soundness and contribution.

Meanwhile, we appreciate the reviewers for pointing out the weaknesses. Your valuable comments help us improve the paper. We try to address each comment as satisfactorily as possible. In the response, we:

a)  compare on a comprehensive benchmarks RULER with 13 sub tasks

b)  discuss some related work such as StreamLLM and induction heads

c)  verify the Desiderata 2 "The similar token gets more attention" empirically on Llama series.

Please find the responses to each reviewer’s comments below. Due to the page limitations, some replies may not be able to provide a detailed description, we welcome any further discussion with the reviewers.

Best regards,

Paper14302 Authors

---

### Decision · Program_Chairs · 2024-09-25

**Decision:**

Accept (poster)

**Comment:**

The paper investigates the role of Rotary Position Embedding (RoPE) in Large Language Models (LLMs), focusing on the relationship between the RoPE base and the model's long-context ability. The study introduces a theoretical lower bound for the RoPE base, validated through empirical experiments. The findings provide actionable insights for practitioners looking to optimize RoPE-based LLMs for longer contexts without superficial gains. The paper highlights potential pitfalls in using small base values, which can lead to a superficial long-context capability.

The reviewers found the paper to be technically sound and impactful, with a strong recommendation for acceptance after addressing the noted weaknesses. The insights provided by this work are expected to benefit the development and optimization of long-context LLMs.